# Selection of Sites for the Treatment and the Final Disposal of Construction and Demolition Waste, Using Two Approaches: An Analysis for Mexico City

**Juan Antonio Araiza-Aguilar** [1,*] , **Constantino Gutiérrez-Palacios** [2],
**María Neftalí Rojas-Valencia** [3], **Hugo Alejandro Nájera-Aguilar** [4],
**Rubén Fernando Gutiérrez-Hernández** [5] **and Rodrigo Antonio Aguilar-Vera** [1]

1   Institute of Geography, National Autonomous University of Mexico, External Circuit, University City, Coyoacan Delegation, Mexico City 04510, Mexico
2   Faculty of Engineering, National Autonomous University of Mexico, External Circuit, University City, Coyoacan Delegation, Mexico City 04510, Mexico
3   Institute of Engineering, National Autonomous University of Mexico, External Circuit, University City, Coyoacan Delegation, Mexico City 04510, Mexico
4   Faculty of Engineering, University of Science and Arts of Chiapas, North Beltway 1150, Lajas Maciel 29000, Tuxtla Gutierrez, Chiapas, Mexico
5   Department of Chemical and Biochemical Engineering, National Technological of Mexico-Technological Institute of Tapachula, Km 2, Highway to Puerto Madero 30700, Tapachula, Chiapas, Mexico
*   Correspondence: juan.araiza@unicach.mx

**Abstract:** This paper proposes a solution to the current problems of Mexico City (Ciudad de México) with respect to construction and demolition waste, through a spatial analysis to locate a waste treatment and disposal infrastructure. Two analysis methodologies, specifically the multi-criteria evaluation technique and network analysis, are used with the support of geographic information systems. The results of the multi-criteria evaluation technique indicate that the most suitable places for this infrastructure location are in the south and southeast of the study area, in the Tlalpan, Milpa Alta, Xochimilco and Cuajimalpa boroughs. The results of the network analysis technique indicate that four facilities strategically located in Miguel Hidalgo, Gustavo A. Madero, Tlahuac and Tlalpan boroughs would permit the provision of service to almost all waste generation points in the study area. Decision makers in Mexico City can use either of the two approaches. If the objective is to find the best location of a single place for the treatment or disposal of huge amounts of waste, the results obtained with the multi-criteria evaluation technique should be used. On the other hand, if waste treatment is favored over final disposal, decision makers should use the results of the network analysis technique.

**Keywords:** construction and demolition waste; location-allocation; multi criteria evaluation; network analyst

## 1. Introduction

The construction sector is an essential part of the world economy, because infrastructure promotes production and employment. Civil works such as buildings, roads, bridges, railways, ports, airports and others, are important sources of capital income and significant urban components [1].

In developed countries or regions such as the UK, the United States, Australia or Hong Kong, the construction sector may account for 4.0 to 10.0% of the Gross Domestic Product (GDP), while in developing countries, it may represent between 3 and 6% [2–5].

In Mexico, this sector represented 6.7% of GDP in 2011, generated around 5.6 million jobs, and positively impacted 63 of the 73 productive branches [6].

Unfortunately, the activities carried out within the construction sector are a source of environmental pollution [7]. Several types of damage are usually caused, such as changes in land use, the pollution of surface waters, generation of greenhouse gases, noise, dust and Construction and Demolition Wastes (CDW) [8–11].

In developed countries, CDW management and their effects have been significantly reduced, mainly due to the development of technical strategies based on reduction, reuse and recycling [12–14]. Regulatory strategies have also been used, for example, International Organization for Standardization (ISO) certifications and environmental control regulations [15–19], as well as management strategies, such as green buildings, within the framework of sustainability [20–22]. However, in developing countries, such as Mexico, the actions taken to manage this type of waste are still incipient.

It is estimated that more than six million tons/year of CDW are generated in Mexico, of which 8.21% comes from Mexico City (MXCD) [6]. However, this is an underestimate, since wastes generated by self-construction, or even those produced by natural phenomena such as earthquakes, are not being accounted for. Currently, the biggest problem facing MXCD is the lack of official sites for waste treatment or final disposal, and few efforts have been made to remedy this situation.

This paper aims to provide a solution to waste treatment and final disposal in MXCD, through an analysis focused on determining the most appropriate locations for CDW treatment and a final disposal infrastructure. Two spatial analysis methodologies are used, with the support of the Geographic Information Systems (GIS), specifically a Multi-Criteria Evaluation (MCE) technique and a network analysis.

The results of this study will be useful for decision making, not only in the study area, but also in other places with similar characteristics. Since MXCD and Mexico (the country) currently lack strategies oriented to determining the appropriate location of an infrastructure for DCW treatment or disposal, the variables taken into account in this paper can be applied in environmental regulation policies. This paper will also offer a theoretical-methodological contribution to solid waste and urban planning, which can be useful to standardize the decision-making process through automated spatial analysis.

## 2. Materials and Methods

### 2.1. Description of the Study Area

Mexico City (MXCD) is one of the 32 states of Mexico (Figure 1). Located within the Valley of Mexico, at an average altitude of 2240 meters above sea level, it has an area of 1495 km$^2$ and is administratively divided into 16 boroughs. Its population of just over 8.9 million inhabitants represents 7.5% of the population of the entire country [23].

For the purpose of regulating land uses and productive activities, MXCD is divided into two zones: Conservation Land (CL) and Urban Land (UL). CL refers to the places that, due to their ecological characteristics, provide the environmental services necessary for the maintenance of the quality of life of the inhabitants. This zone covers approximately 87.297 Ha (58.39% of MXCD), and is located in the south and south west of the study area [24,25]. On the other hand, UL corresponds to the central and northern part of MXCD, where the land has been ascribed to various uses, such as residence, equipment, offices, industrial parks and others. UL occupies the rest of the city area, i.e., 41.61% [26].

### 2.2. CDW Management in MXCD

In MXCD, the Construction and Demolition Wastes (CDW) generation rate is currently unknown, but some studies have been conducted to try to determine it, and one of them suggests that more than 10,000 tons/day are generated (2019), particularly in the aftermath of a devastating earthquake in September 2017 [27].

From generation to final disposal, CDW follows various routes. The main CDW sources include public works carried out by the government of MXCD (35%), private works performed by different construction companies (50%), and self-construction (15%). The generation rates by source vary considerably, but there is a slight predominance of waste generated by private works. Additionally, in 2017 an earthquake of magnitude 7.1 occurred, which also generated a large amount of CDW in various parts of MXCD (see Figure 1); It is estimated that the amount of waste generated by demolished houses was 46,708 m$^3$, equivalent to 86,409 tons [28].

CDW collection is mainly conducted by private collection services (76.5%) that are controlled by the Secretariat of Environment (SEDEMA), particularly through a registry called RAMIR [29]. An important percentage of CDW (20%) is illegally collected by small contractors. Finally, municipal solid waste collection trucks collect the remaining part of this waste, especially when it is mixed. This can be seen in several of the waste generation inventories of MXCD, where the CDW generation rates within the total mix range from 1 to 6% (averaging 3.5% for the years 2010 to 2016) [30–35].

After collection, between 3 and 10% of the CDW is sent for recycling, particularly to a plant called "Concretos Reciclados" ("Recycled Concrete"), which converts CDW into construction raw materials. A relevant part of the CDW is sent to 20 final disposal sites that have been authorized by SEDEMA [36]. However, all of them are located outside of the study area, and lack appropriate engineering measures for final disposal. Another significant amount of the waste generated is deposited in clandestine dumpsites throughout MXCD, causing negative aesthetic impacts, according to Procuraduría Ambiental y del Ordenamiento Territorial (PAOT) [37]. All the information given above can be seen in Figures 1 and 2.

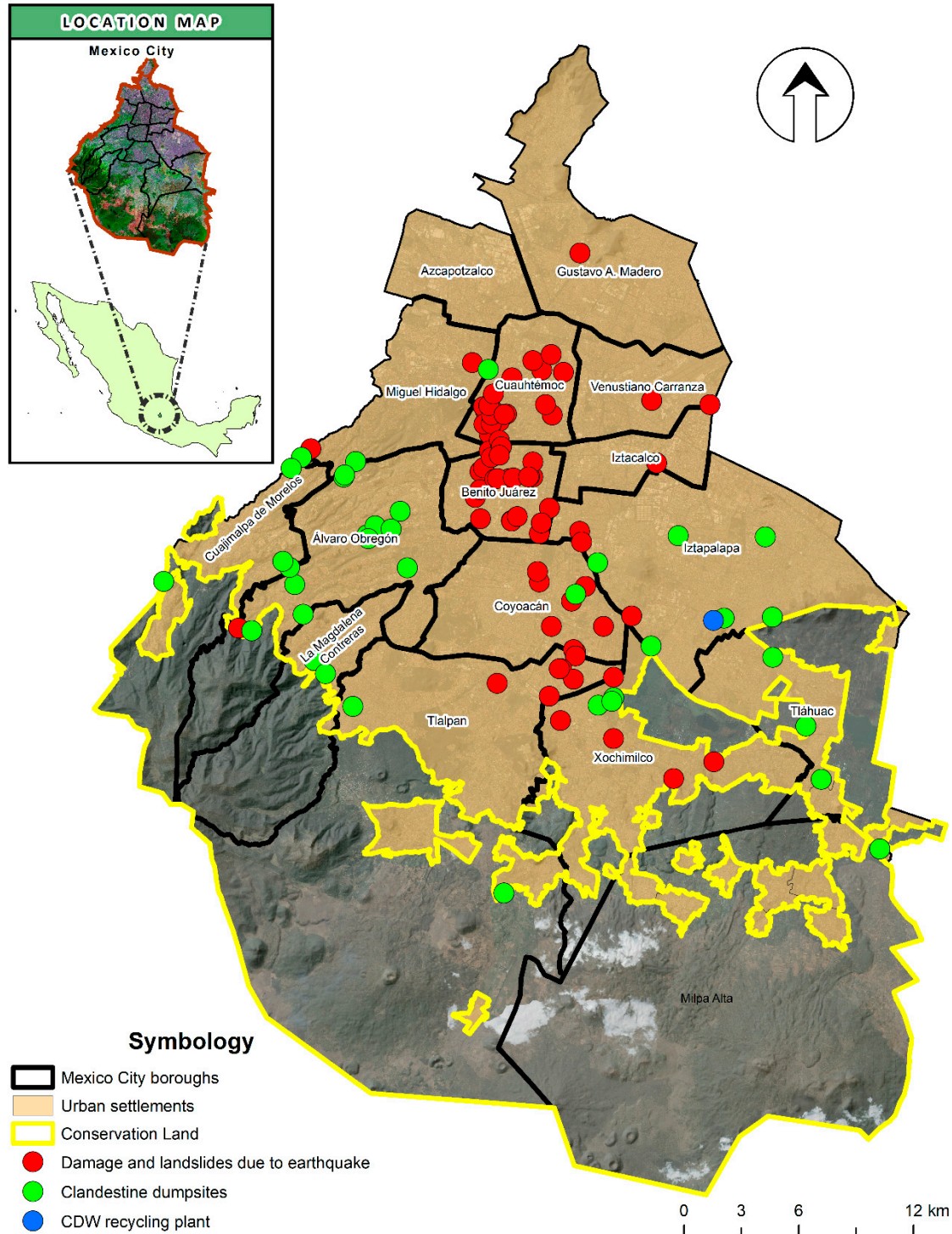

**Figure 1.** Location of the study area.

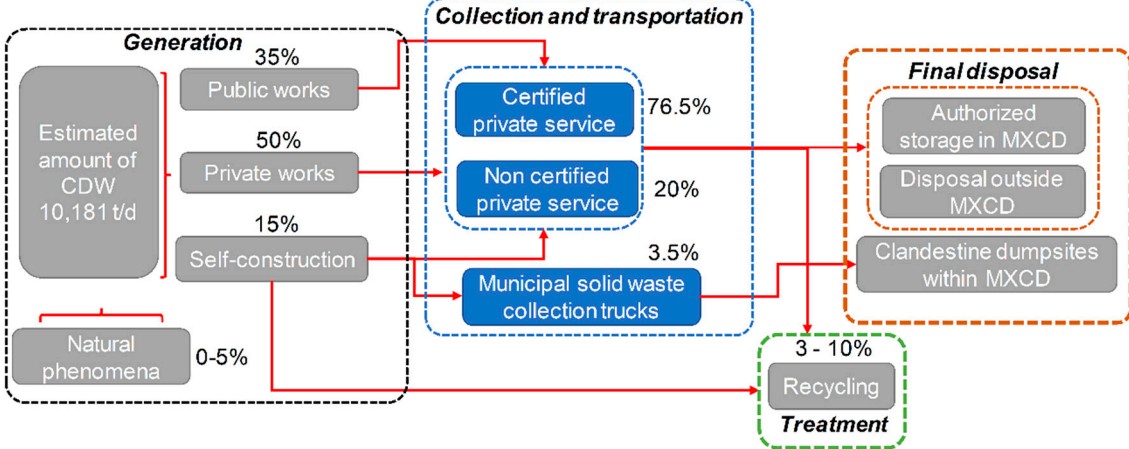

**Figure 2.** Diagram of Construction and Demolition Wastes (CDW) management in Mexico City (MXCD).

*2.3. Spatial Analysis Methodologies*

In this paper, two methodologies based on Geographic Information Systems (GIS) were used to determine the best location of possible CDW treatment and final disposal sites within MXCD. The first emplacement approach was based on the use of a Multi-Criteria Evaluation (MCE) technique, which allowed us to analyze this problem under the hypothesis of the "capacity of reception of the territory", as described by Gómez and Barredo [38]. The second emplacement approach was a network analysis, specifically "location-allocation", which allowed us to choose the best locations out of a set of candidate facilities, which operate properly (short routes, shorter times) as a function of their potential interaction with the demand points (CDW generation sites).

2.3.1. Emplacement by MCE

The MCE spatial analysis technique used in this work was the Weighted Linear Combination (WLC), with the suggestions given by Eastman et al. [39] and Malczewski [40], especially the incorporation of some restrictive variables within the structure of the equation used (Equation 1). In that equation, $S$ is the suitability index; $w_i$ is the weight of the selection variable $i$; $x_i$ is the score of the selection variable $i$; $C_i$ is the score of the restriction variable $j$, and $\prod 0$ is the product of them.

$$S = \left( \sum_{i=1}^{n} w_i \, x_i \right) \left( \prod C_{ij} \right) \tag{1}$$

EMC techniques have been widely used with good results in solid waste management, for example, to locate landfills [41,42], transfer stations [43,44], or treatment sites [45,46].

(A) Delimitation of the area of analysis for locating waste handling infrastructure by MCE

An area of analysis was delimited within the study area, which is large and mostly covered by human settlements, where the required infrastructure could not be established. Figure 3 shows the delimitation of the area of analysis, which is located mainly in the southern part of MXCD, along the border between Conservation Land (CL) and Urban Land (UL). Additionally, other smaller areas were also delimited in the north, east and west parts of the city. Note that this area of analysis also worked as an emplacement variable for the required infrastructure.

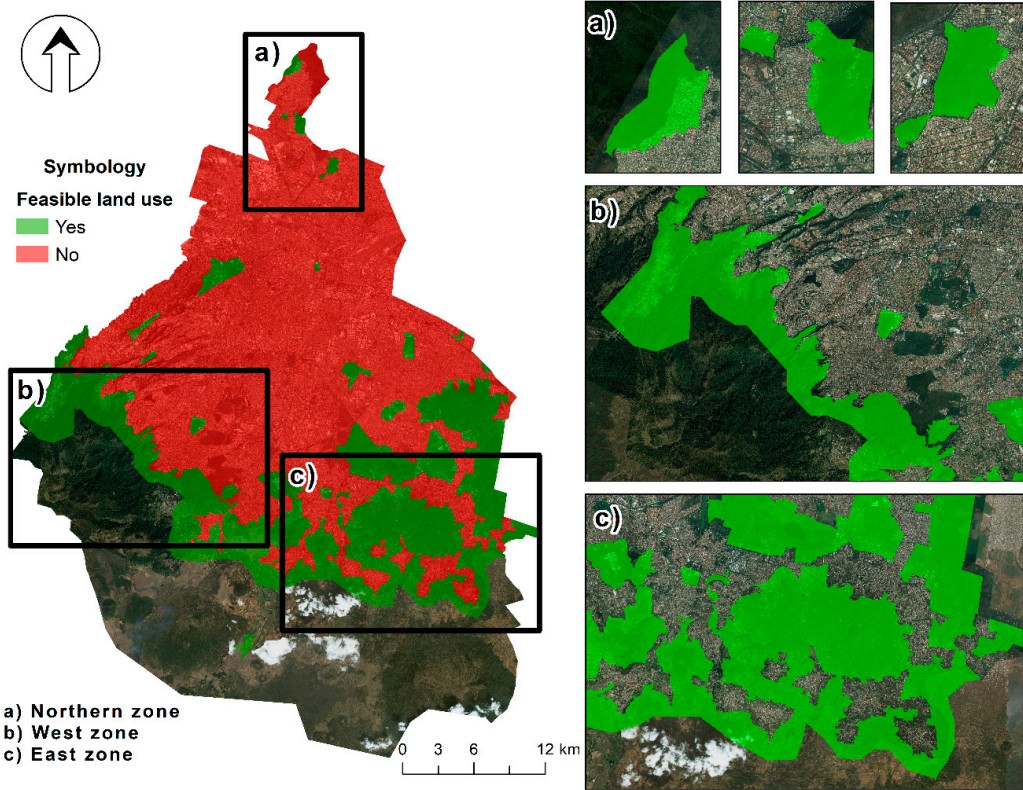

**Figure 3.** Delimitation of the area of analysis for locating a waste handling infrastructure by Multi-Criteria Evaluation (MCE).

(B) Definition of emplacement variables

In various papers, UNAM [27], Banias et al. [47] and Dosal et al. [48] have proposed several variables to evaluate the technical, economic and environmental feasibility of CDW treatment plants and final disposal sites. The variables used are usually "the topography or slopes, the distances with respect to other infrastructures, transport costs, land value, or even damage caused by the emissions of dust or noise". However, many of these variables, although important, cannot be used in this first emplacement approach, since they do not have a spatial nature, and their spatial construction would require much time and effort.

For this work, variables that are important and accessible for the study area, especially from the geological and people safety viewpoints, have been selected. The emplacement variables used were: (i) Slopes, (ii) flood zones, (iii) landslide zones and (iv) geotechnical zoning. On the other hand, the restriction variables used in this work were: (i) Areas of natural relevance, such as protected natural areas, or areas of high environmental value and (ii) uninhabited areas.

(C) Variable weighting

In this work, the paired comparison by Saaty [49] was used to weight the emplacement variables. This type of analysis does not apply to restrictive variables, since only one product function (multiplication) is used with them. Table 1 shows a pairwise comparison matrix, in which all of the grouped variables are found, in order to compare them with respect to the others; in this way, the relative importance of each of them is analyzed and then the weights are obtained. Paired comparison is usually done with the help of a reference scale, which ranges from 9, 8, 7, to 1, including reciprocal ones, where each number means the number of times that one variable is more important than another [50].

It should be noted that the main objective of this matrix is to obtain the weights of each variable, which will then be used within Equation (1). It is important to mention that, prior to the use of the weights, it is necessary to analyze the logical consistency of the matrix. This is done through the

Consistency Ratio (*CR*) and the Consistency Index (*CI*), which are calculated by means of equations 2 and 3. In these Equations $\lambda_{max}$ is the maximum eigenvalue obtained from Table 1, $RI_n$ is a random index that is obtained by the means of tables, and $n$ is the order of the matrix used.

$$CI = \frac{(\lambda_{max} - n)}{(n - 1)} \tag{2}$$

$$CR = \frac{CI}{RI_n} \tag{3}$$

According to Saaty [49], matrices with values of $CR \leq 0.1$ must be accepted, and larger values rejected. A value greater than 0.1 means that any judgments are 10% as inconsistent as if they were random. This situation can cause adjustments in the matrix through the corrections of the importance values of each variable.

**Table 1.** Pairwise comparison matrix.

|  | Variables | i.1 | i.2 | i.3 | i.4 | Sum | Weight |
|---|---|---|---|---|---|---|---|
| **i.1** | Geotechnical zoning | 1.00 | 2.00 | 3.00 | 5.00 | 11.00 | 0.51 |
| **i.2** | Flood zones | 0.50 | 1.00 | 1.00 | 2.00 | 4.50 | 0.21 |
| **i.3** | Landslide zones | 0.33 | 1.00 | 1.00 | 1.00 | 3.33 | 0.15 |
| **i.5** | Slopes | 0.20 | 0.50 | 1.00 | 1.00 | 2.70 | 0.13 |
| **Total** |  | 2.03 | 4.50 | 6.00 | 9.00 | 21.53 | 1.00 |

$\lambda_{max} = 4.04$, $CI = 0.012$, $RI_n = 0.89$ and $CR = 0.014 < 0.1$

(D) Normalization of levels by variable

In order to normalize the levels of each emplacement variable, a simple assessment was used with values ranging from 1 to 3, where the smaller value corresponds to the most unfavorable condition, while the highest value corresponds to the most favorable condition. Restrictive variables having Boolean values of 1 and 0 are used. Therefore, by multiplying them, a layer with ideal territorial surface will be obtained to locate the infrastructure. In this way, the territorial areas where it is not possible to place absolutely anything, are also eliminated. The detailed information on the values adopted for each variable is shown in Table 2.

**Table 2.** Normalization levels for each variable used.

| Variables | Levels | Normalized Level |
|---|---|---|
| Geotechnical zoning | Zone I | 3 |
|  | Zone II | 2 |
|  | Zone III | 1 |
| Flood zones | Outside flood zones | 3 |
|  | Within flood zones | 1 |
| Landslide zones | Outside areas of landslides | 3 |
|  | Within areas of landslides | 1 |
| Slopes | <5° | 3 |
|  | 5–25° | 2 |
|  | >25° | 1 |

### 2.3.2. Emplacement by network analysis

The location of service infrastructure through location-allocation models began to be used in the 1970s, especially with the appearance of the first spatial databases [51]. In solid waste management, these techniques have been used to analyze infrastructure, such as containers or transfer stations along

a transport network [52–55]. The solution to these problems is based on a combinatorial analysis (Equation (4)), where given *N* candidate facilities and *M* demand points with a weight, a subset of facilities *P* must be chosen (where *P* < *N*); such that the sum of the weighted distances from each *M* to the closest *P* is minimized.

The location-allocation is a twofold problem, because it simultaneously locates facilities and allocates demand points to those facilities [56]. Unfortunately, although simplifications are made, the results of the combinatorial analyses are enormous, and for this reason it is common to use heuristic mechanisms, based upon trial and error procedures towards a continuous approach to the best solution [57].

$$\binom{N}{P} = \frac{N!}{P!(N-P)!} \tag{4}$$

The use of location-allocation models involves the knowledge of the study area and the operation of the infrastructure to be installed. Church [58] and Buzai [59] have proposed a way to select these models, based on the nature of the service provided by the facility. For example, if the service is private, models that improve spatial efficiency should be used. The existence of other competitors in the study area must also be considered. If the service is public, models that try to improve spatial equity should be used.

(A) Delimitation of the area of analysis for locating waste handling infrastructure by network analysis

Figure 4a shows the area of analysis for locating the waste handling infrastructure by network analysis. This area is located in the central zone of MXCD, especially where connection to a transport network is available (Figure 4b). The south zone of the City is not taken into account, since this territorial surface is within CL. Moreover, relatively few human settlements are located there, and CDW generation is almost nil. It should be noted that, from the MCE results, several candidate facilities were identified.

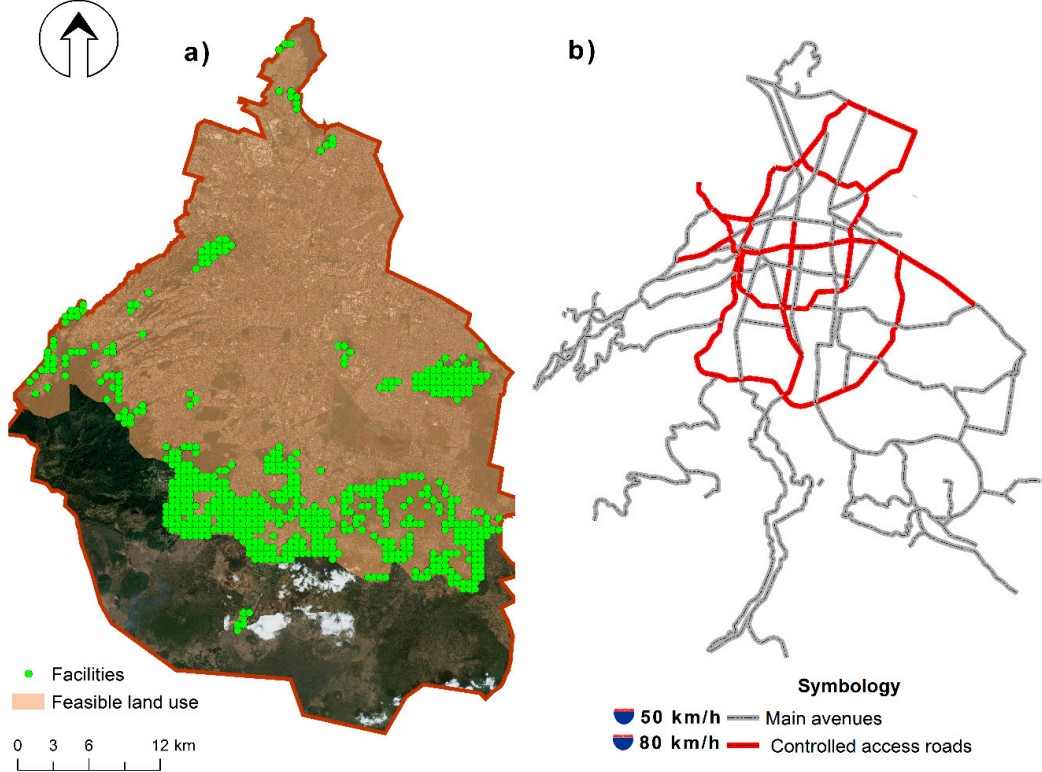

**Figure 4.** Delimitation of the area of analysis for locating waste handling infrastructure by network analysis: (**a**) area of analysis within feasible land use; (**b**) transport network.

(B) Primary components of the analysis

The location-allocation models takes into account three primary components: (i) Demand points, (ii) candidate facilities, and (iii) a distance-time matrix. In this paper, the demand points are the places where CDW is generated, for example, the centroids of population zones (2432 points), the location points of buildings collapsed by earthquakes (74 points), and the location points of clandestine CDW dump sites (36 points).

On the other hand, the candidate facilities are those shown in Figure 4b, i.e., centroids located in different areas of MXCD, which meet the emplacement criteria, especially the reception capacity of the territory (605 points). Finally, the distance-time matrix is used to analyze the distances or travel time between the candidate facilities and demand points. It was built using ArcGis 10.X.

(C) Construction of transport networks

The GIS database of the study area was built with the support of the ArcGis 10.X software, with information from INEGI [60] and SCC [61]. This database contains several types of feature classes (line-point or arc-nodes), such as primary roads and controlled access roads. It includes 200 nodes and 262 arcs, the latter with a total length of 650 km. In order to eliminate possible errors, this database was additionally reviewed through a topological analysis.

The length of each arc was determined by the software used, through the "Calculate Geometry" function. Equation (5) was used to obtain transport times, where *FT or TF minutes* corresponds to the transport time in both directions of the analyzed arc; *SHAPE_Length* is the length of the arc analyzed in meters; *FT or TF_Speed* is the speed allowed in both directions of the arc analyzed in km/h; finally, 0.06 is the conversion factor to obtain these transport times in minutes.

$$FT \ or \ TF \ minutes = \frac{0.06 \ xSHAPE\_Lenght \ (m)}{FT \ or \ TF\_Speed \ (km/h)} \tag{5}$$

(D) Location-allocation problems

Through the software ArcGis version 10.X and some premises or input data, seven types of location-allocation problems can be solved. These problems are: (i) To minimize impedance; (ii) maximize coverage; (iii) minimize facilities; (iv) maximize attendance; (v) maximize capacitated coverage; (vi) maximize market share; and (vii) target market share. In this paper, the first four problems were solved. The last three problems were not included, since they are frequently used to solve competitive problems of the location of facilities. A brief description of each location-allocation problem is presented below:

Minimize impedance: This location-allocation problem attempts to determine the appropriate locations for a series of candidate facilities, in order to minimize the sum of all weighted costs between the demand points and the facilities. This model, often called "P-Median Problem", was developed by Hakimi [62,63]. Its objective function according to Aremu et al. [54] and Miller and Shaw [64] is as follows:

$$\text{Minimize } Z = \sum_{i=1}^{n} \sum_{j=1}^{m} c_{ij} x_{ij} \tag{6}$$

Subject to the following restrictions:

$$\sum_{j=1}^{m} x_{ij} = 1, \qquad i = 1, \dots, n \tag{7}$$

$$x_{jj} - x_{ij} \geq 0, \qquad i = 1, \dots, n; \qquad j = 1, \dots, m; \qquad i \neq j \tag{8}$$

$$\sum_{j=1}^{m} x_{jj} = P \tag{9}$$

where $\{1,\ldots,n\}$ is the set of demand points; $\{1,\ldots,m\}$ is the set of potential points where the facilities could be located; $c_{ij}$ is the distance between the demand point $i$ and the candidate facility $j$; $P$ is the number of facilities to be located; finally $x_{ij}$ and $x_{jj}$ are Boolean variables, defined as:

$$x_{ij} = \begin{cases} 1 & \text{if demand point } i \text{ is allocated to facility at } j \\ 0 & \text{Otherwise} \end{cases}$$

$$x_{jj} = \begin{cases} 1 & \text{if a facility is located at site } j \\ 0 & \text{Otherwise} \end{cases}$$

Equation (6) minimizes the total distance between all of the demand points and their assigned facilities. Equation (7) guarantees that each demand point is allocated only to the nearest facility. Equation (8) states that a waste generator can only be allocated to a facility that has been located. Finally, Equation (9) ensures that the number of facilities located equals the number of facilities allocated.

Maximize coverage: This model chooses facilities so that the maximum number of demand points are covered (attended) within a specified impedance value, which can be the distance or time of service [56]. Church and Reveille [65] developed this location-allocation problem, and its objective function is as follows:

$$\text{Maximize } Z = \sum_{i \in I} a_i y_i \tag{10}$$

Subject to the following restrictions:

$$\sum_{j \in N_i} x_j \geq y_i \text{ for all } i \in I \tag{11}$$

$$\sum_{j \in J} x_j = P \tag{12}$$

where $a_i$ is the quantity of demand at point $i$; $I$ is the set of demand points; $J$ is the set of candidate facilities; $P$ is the number of facilities to be located; $N_i$ is the set of facilities capable of covering the demand of point $i$. Finally, the variables $x_j$ and $y_i$ are defined by the following:

$$x_j = \begin{cases} 1 & \text{if a facility is located at site } j \\ 0 & \text{Otherwise} \end{cases}$$

$$y_i = \begin{cases} 1 & \text{if the demand point } i \text{ is convered by a facility within the specified impedance S} \\ 0 & \text{Otherwise} \end{cases}$$

$$S = \text{is the distance or the maximum service time}$$

Equation (10) maximizes the number of covered demand points within a specified impedance value. Equation (11) guarantees that demand point $i$ is allocated to a selected facility, and ensures that all facilities that are allocated to demand point $i$ are located within the specified distance or time limit. Equation (12) indicates that there are $P$ facilities to be located.

Minimize facilities: This location-allocation problem is similar to the problem of maximizing coverage, with the exception that the number of facilities to be located is determined by the software used, which in this case is ArcGis 10.X [56]. The minimize facilities model has received multiple contributions from researchers who study the emplacement of public and private services [66–68]. Its objective function is as follows:

$$\text{Minimize } Z = \sum_{j=1}^{m} x_j \tag{13}$$

Subject to the following restrictions:

$$\sum_{j \in N_i} x_j = 1 \qquad i = 1, \ldots, n \qquad (14)$$

In Equations (13) and (14), $N_i$ is the set of candidate facilities located within the distance limit, and that can attend the demand point $i$ ($\{j | c_{ij} \leq S\}$); $c_{ij}$ is the distance between the demand point $i$ and the candidate facility $j$; $S$ is the distance or the maximum time of service; $x_j$ is a Boolean variable defined similarly to the previous cases.

Maximize attendance: Holmes et al. [69] developed this location-allocation problem, which seeks to maximize the number of demand points that a facility can attend at a specific impedance limit. This maximization problem assumes that the interaction between the locations of the facilities and the locations of demand points decreases as the distance increases, i.e., the optimal locations for the facilities will be located very close to the greatest number of demand points (Equation (15)). The objective function of this location-allocation problem is the following:

$$\text{Maximize } Z = \sum_{i=1}^{n} \sum_{j=1}^{n} a_i \left( S - c_{ij} \right) x_{ij} \qquad (15)$$

where $a_i$ is the quantity of demand at point $i$; $S$ is the distance or the maximum time of service; $c_{ij}$ is the distance between the demand point $i$ and the candidate installation $j$; $x_j$ is a Boolean variable defined similarly to the previous cases.

(E) Impedances and other input data in location-allocation problems

Impedance is the specific property that indicates the cost of traveling along a network, which may be time or distance. In this paper, to determine the location of the CDW treatment and the final disposal infrastructure, it was carried out considering only the distance as the impedance factor, with a cut-off value of 20 km. In addition, the calculation of the displacement costs was determined only in the sense of transportation from demand points to facilities points.

Only four optimal facilities were considered to handle all CDW generation points in MXCD, except as regards the problem "minimize installations", since the software solver that was used determined the minimum number of installations. Finally, the considered impedance decay or transformation was linear (value = 1).

## 3. Results and Discussion

### 3.1. MCE Approach

The MCE emplacement approach focuses on locations allowing a permanently operating infrastructure giving service to a large territorial extension, so that large amounts of CDW can be treated and deposited in one place. Suitable locations determined by this emplacement approach cannot be within any urban settlements, except for a few places that meet the selection requirements.

Figure 5a shows the variable "slope", which is very important because it is related to the vertical extension or height of the deposit cells, and their stability. This variable also influences the degree of excavation and leveling of the land where the CDW treatment and final disposal infrastructure is located. Figure 5b–d are the variables "landslide zones, flood zones and geotechnical zoning" respectively, which are related to the safety of the potential infrastructure and nearby settlements. An incorrect location with respect to these variables can cause the CDW to be unstable and dragged, potentially generating severe damage.

Boolean or restrictive variables work as a filter to eliminate those territorial areas where it is not possible to locate any type of infrastructure. The variable called "area of natural relevance" (Figure 5f) is very important for MXCD, since places with a high environmental value are abundant, and strictly protected by federal and state regulations. Moreover, these places host a large number of animal and

vegetables species. The feasible land use variable (Figure 5g) aims to find territorial areas where land use is compatible with waste deposit, and where population is least affected.

Figure 5e,h show the combination of the restrictive and site variables, while Figure 5i shows the final result of the MCE technique in the GIS environment. The area of analysis is classified into three categories, based upon the degree of suitability to locate the CDW treatment infrastructure and the final disposal site. The largest territorial area (149.46 km²), located mainly in the south and southeast of the area of analysis, in the "Tlalpan, Milpa Alta and Xochimilco" boroughs, belongs to the "high suitability" category.

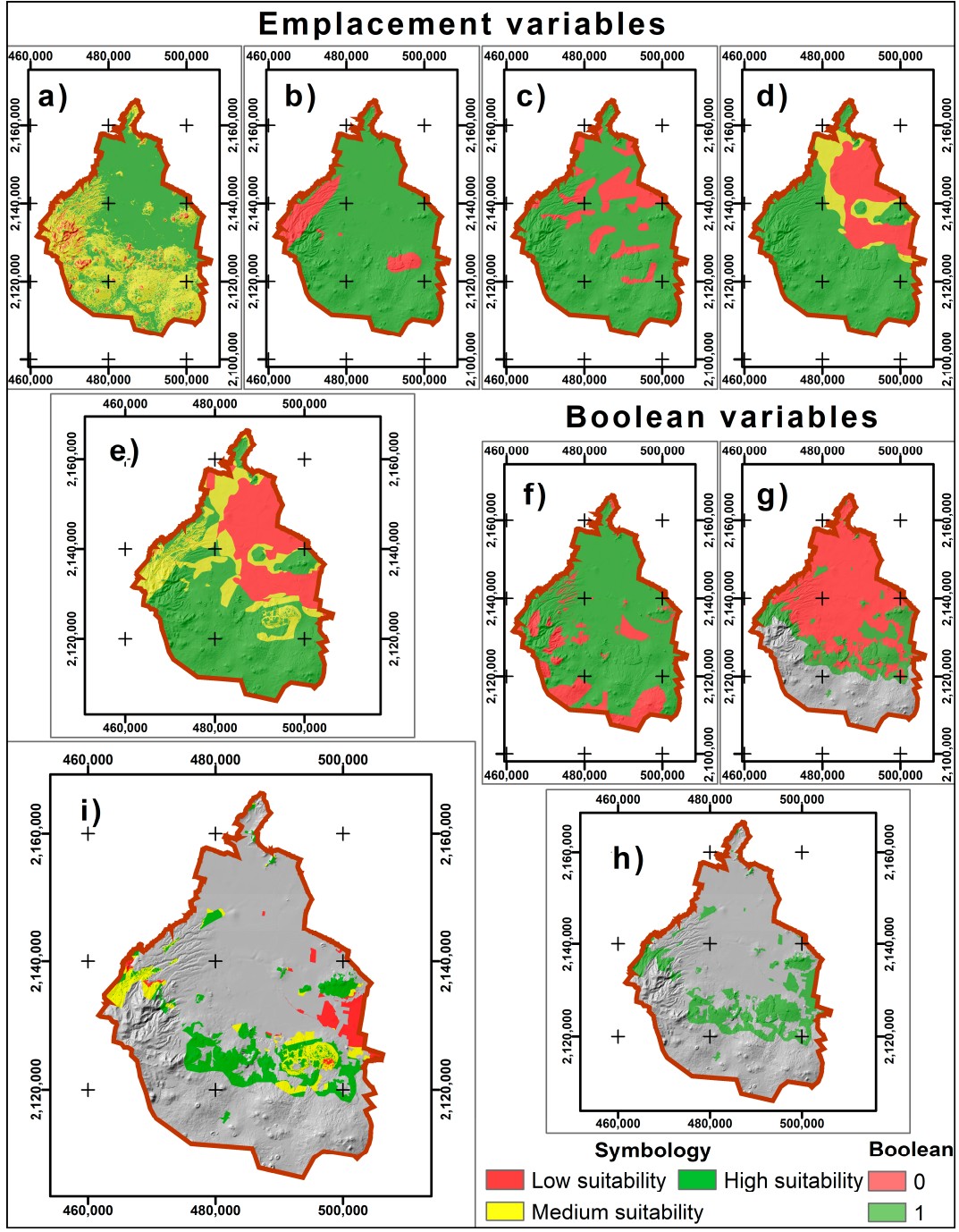

**Figure 5.** Emplacement by MCE analysis: (**a**) slopes; (**b**) landslide zones; (**c**) flood zones; (**d**) geotechnical zoning; (**e**) combination of emplacement variables; (**f**) area of natural relevance; (**g**) feasible land use; (**h**) combination of restrictive variables; (**i**) final map of MCE technique.

The area classified as "medium suitability" covers 66.83 km², and is located in the southeast and west of the area of analysis, in the "Milpa Alta, Cuajimalpa and Xochimilco" boroughs. Finally, 41.08 km² belong to the "low suitability" category, and should not be considered, since many of the evaluated variables present their most unfavorable condition. Figure 6 shows the surface distribution of each category of suitability.

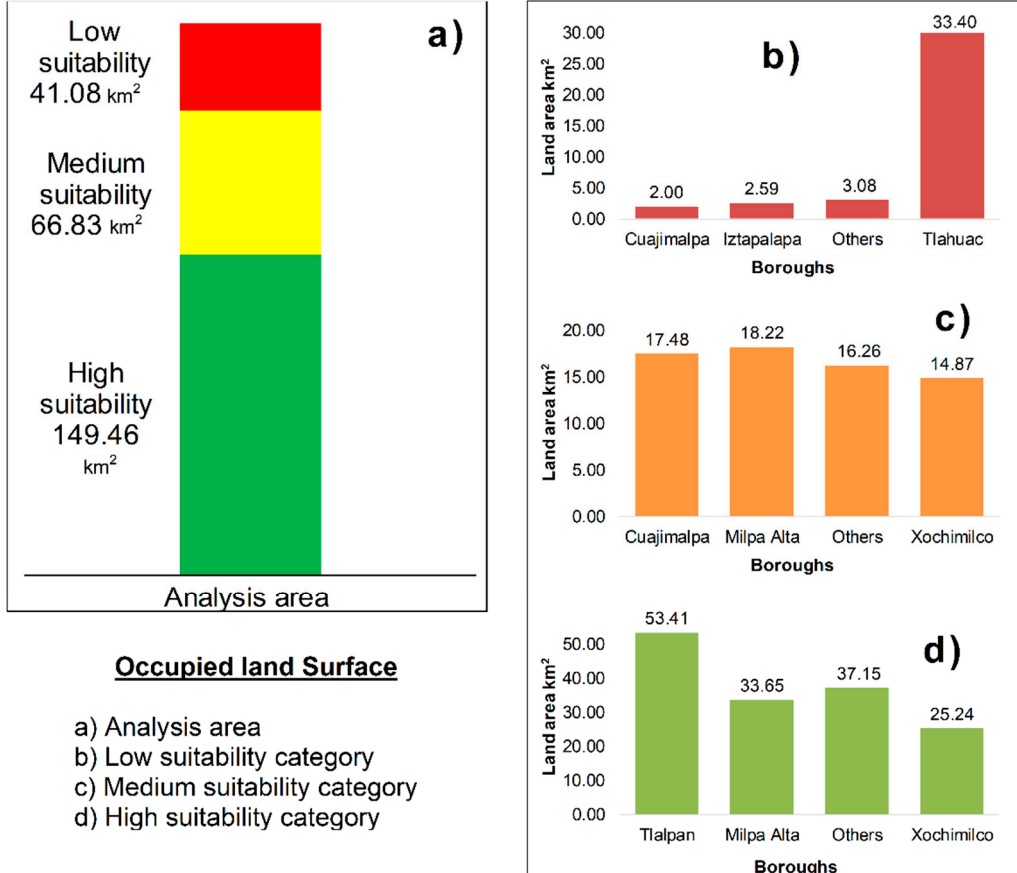

**Figure 6.** Distribution of occupation by suitability category.

### 3.2. Network Analysis Approach

The object of the network analysis (location-allocation) approach is to choose the best infrastructure locations based on their interaction with the demand points. Unlike the MCE approach, it allows the infrastructure to be located within urban settlements. However, its surface area must be small, and its operation must be more oriented to waste treatment than to final disposal.

The results of the different location-allocation problems are shown in Figures 7 and 8. Minimizing impedance and maximizing coverage (Figures 7a and 8a,b) present similar results, since the four facilities chosen are located in the same places, in the Miguel Hidalgo, Gustavo A. Madero, Tlahuac and Tlalpan boroughs. The four chosen facilities give service to the same number of demand points (2510 points), leaving only 32 unattended. Moreover, these facilities offer the shortest travel distances.

The problem of maximizing attendance (Figures 7b and 8c) identified four facilities chosen in the Miguel Hidalgo, Gustavo A. Madero, Iztapalapa and Tlalpan boroughs. The proposed solution serves 2475 demand points, leaving 67 unattended. This location-allocation problem offers results that may also be considered acceptable.

Finally, the problem of minimizing facilities (Figures 7c and 8d) shows that the minimum facilities to give service to almost all of the demand points should be eight instead of four, with locations in Miguel Hidalgo, Gustavo A Madero, Iztapalapa, Tlalpan (three facilities) and Xochimilco (two facilities) boroughs. With the solution to this location-allocation problem, 2537 demand points are served, leaving only 5 unattended. Additionally, this option handles more demand points than the others; however, its operation and construction would be very expensive, since three of the chosen facilities would only serve 100 or fewer demand points.

On the other hand, it should be noted that none of the location-allocation problems modeled in this paper are considering the CDW reception capacity in each chosen installation. It will therefore be indispensable that MXCD decision makers include this new variable in future modeling, so as to obtain more accurate results. This new variable can be modeled from the problem called "maximize capacitated coverage".

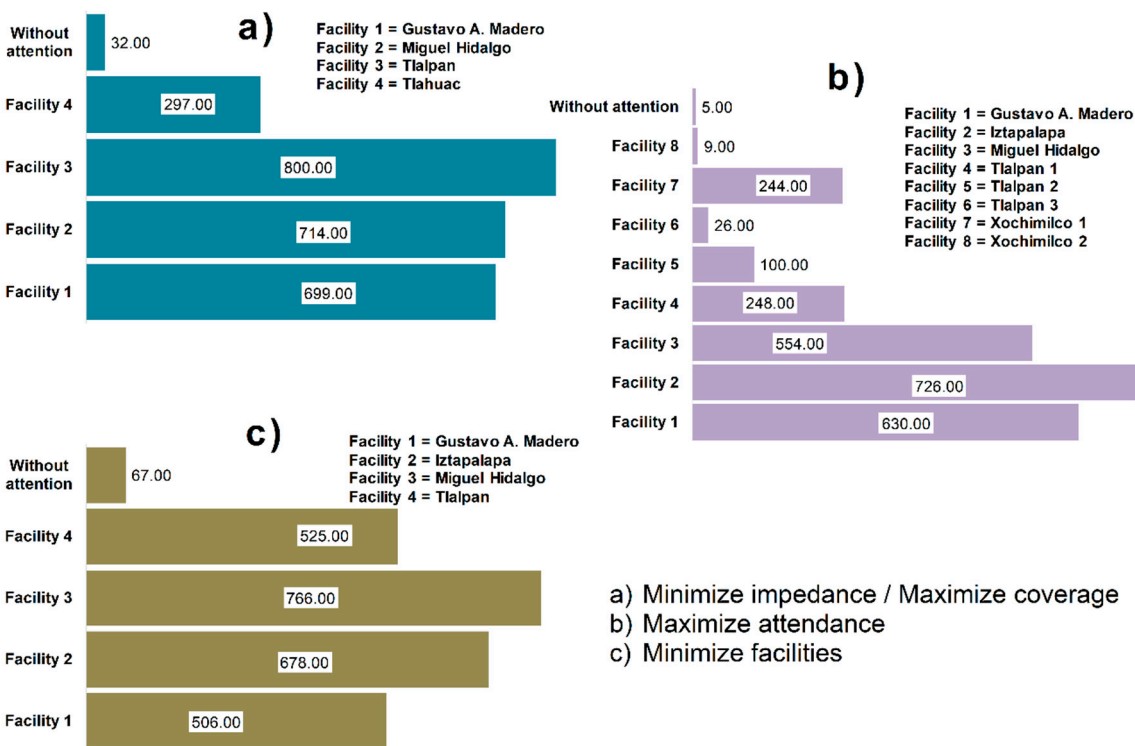

**Figure 7.** Demand points served by the chosen facilities.

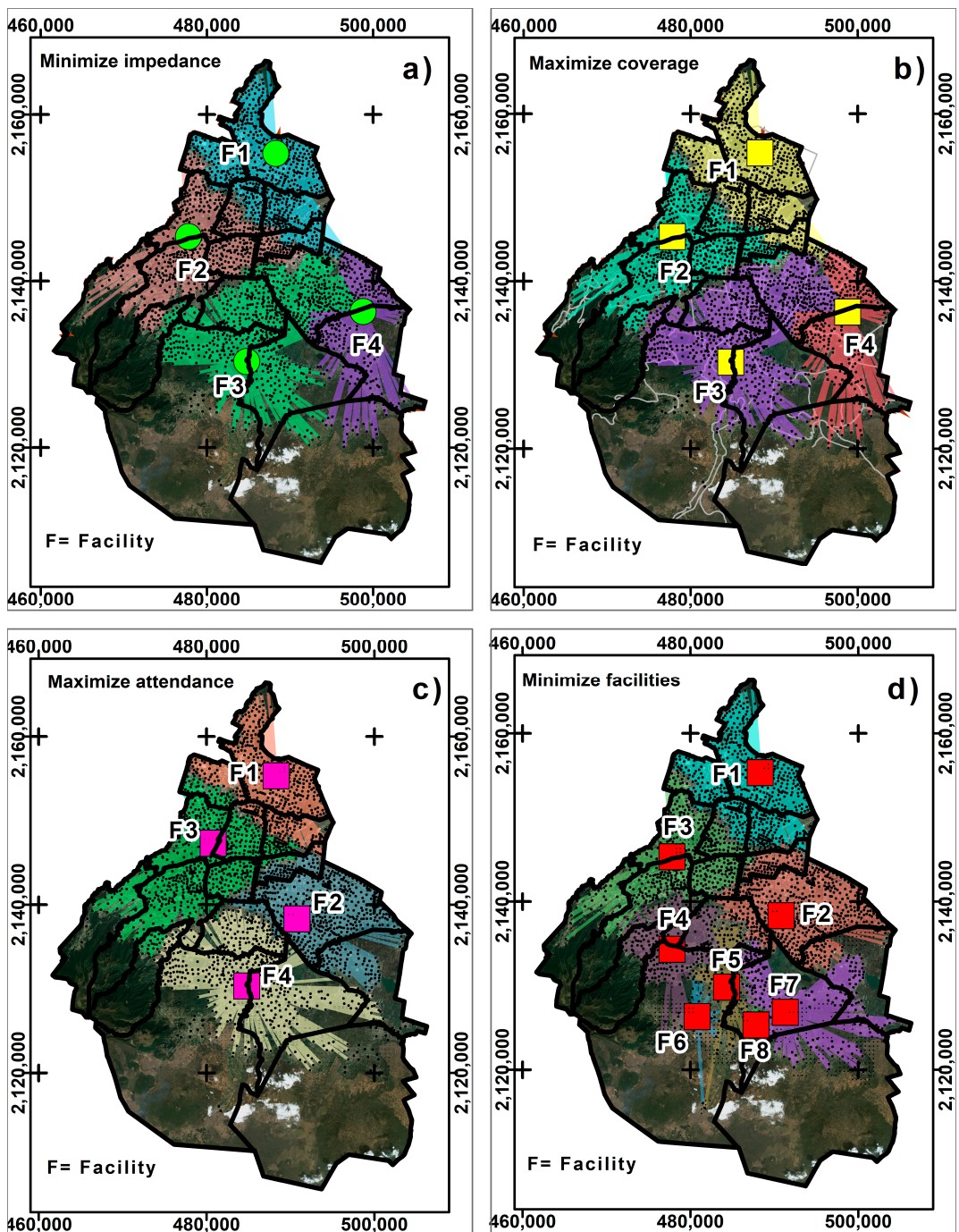

**Figure 8.** Emplacement by network analysis (location-allocation): (**a**) minimizing impedance problem; (**b**) maximizing coverage problem; (**c**) maximizing attendance problem; (**d**) minimizing facilities problem.

### 3.3. Final Considerations

Nowadays, it is difficult to determine the optimum location of facilities because this process involves several variables and actors [46]. Regulations focused on infrastructure location, for example, NOM-083-SEMARNAT-2003 [70] and NTEA-010-SMA-RS-2008 [71], are available in Mexico, but none of them is applicable to CDW handling. Therefore, the spatial techniques and the results of this study are an important theoretical-methodological contribution, which can be used as a basis to begin formulating these regulations.

Independently of an appropriate location determined through space techniques, a correct operation of CDW treatment infrastructure and final disposal must take into account regulatory policies. For example, regulatory and management strategies, such as ISO certifications and green constructions, can make civil work building processes more efficient and improve the flow of waste, preventing thus inappropriate disposal. Technical strategies based on reuse and recycling can also help reduce the volume of CDW arriving at the treatment plants or final disposal sites.

Other agencies in the construction sector can carry out activities to improve regulatory policies and CDW management. For example, the Chambers of the Construction Industry, through their members, can create intensive awareness campaigns to prevent incorrect CDW handling in civil works, and can develop technical studies to quantify CDW, and then ascertain how and where to dispose of them properly. This study can be a useful basis for these purposes.

## 4. Conclusions

This paper proposed a solution to the CDW handling problem in MXCD, through a spatial analysis to determine where to best locate CDW treatment and final disposal infrastructure. Two analysis methodologies were used with GIS support, specifically a network analysis and an MCE technique.

MXCD decision makers can choose between two approaches to best locate the infrastructure. If they wish to establish a single place where large amounts of waste are treated or disposed of, they should use the spatial modeling offered by the MCE technique. The results of this technique indicate that the most suitable area is found in the south and southeast of the area of the analysis, in Tlalpan, Milpa Alta, Xochimilco and Cuajimalpa boroughs.

On the other hand, if the decision makers wish to establish an infrastructure more oriented towards CDW treatment than final disposal, they should use the spatial modeling offered by network analysis. The results of this technique, particularly the solution of problems to minimize impedance and maximize coverage, indicate that four strategically located facilities are sufficient to give service to almost all CDW generation points. These facilities should be located in Miguel Hidalgo, Gustavo A. Madero, Tlahuac and Tlalpan boroughs.

It is important to note that both options have pros and cons related to cost, operation and the spatial location of each type of infrastructure. For this reason, it is essential that MXCD decision makers carefully analyze the future operation of the CDW treatment and the final disposal plant.

The results obtained in this study were satisfactory, since they permitted to answer important questions in the fields of solid waste and urban planning, such as where CDW treatment facilities or final disposal should be located, and how many facilities would be necessary to give service to all CDW generation points in a given area.

Due to the limited information currently available on the operation of the facilities, other questions may be answered in future studies, such as how the suitability areas are adjusted, by using social or economic variables, or how space is efficiency modified, by including data on the reception capacity of each chosen facility.

**Author Contributions:** J.A.A.A. and R.A.A.V.—conceived and designed the methodology, collected and analyzed the data; H.A.N.A. and R.F.G.H.—wrote the paper; C.G.P. and M.N.R.V—conceptualized, revised and edited the paper; C.G.P.—provided the funding.

**Funding:** This work was supported by the Secretariat of Science, Technology and Innovation of Mexico City.

**Acknowledgments:** The preparation of this paper was possible thanks to the support of the Secretariat of Science, Technology and Innovation of Mexico City, through the agreement SECITI/090/2018. We also thank the reviewers for their useful criticisms and suggestions.

**Conflicts of Interest:** The authors declare no conflict of interest.

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
