# Peer review of "Selection of Sites for the Treatment and the Final Disposal of Construction and Demolition Waste, Using Two Approaches: An Analysis for Mexico City"

_sustainability, doi:10.3390/su11154077_

Round 1
Reviewer 1 Report
This article presents two quantitative methods for identifying sites for construction & demolition waste treatment and disposal in Mexico City. Site selection remains an important problem, especially in densely settled regions. Overall the article is well constructed and well-written, but a few aspects need more development:
1. The most significant problem is that the manuscript ignores the issue of C&D reuse and recycling. I understand that this is not the focus of the research; however many cities have had great success in reducing C&D waste through recycling, and mandated use of building techniques that limit waste generation. The authors should at least acknowledge this policy possibility as it would significantly reduce the need for c&d disposal and thus reduce the need for new C&D disposal infrastructure.
2. Though the comparison of the MCE and network methods was interesting, the authors have limited their inputs, perhaps too much. The highly technocratic approach fails to consider impacts on neighboring residents or indirect environmental impacts, focusing only of the efficiency of transport and disposal, direct environmental impacts, and conditions relevant to waste facilities themselves. Also, beyond acknowledging conservation areas where development is prohibited, the authors have not acknowledged or explored any other regulatory contexts that could be relevant. Though an important investigation, the highly limited and technocratic approach to siting seems a bit outdated, particularly in an environmentally sensitive context like MXCD.
3. Finally, the authors could do more to generalize their finding. As it is currently written, the article is relevant only to C&D waste policy makers, within MXCD. In order to be published, the authors should consider how the methodological approach or the findings are relevant to policy-makers in other places, or other sectors.
Author Response
First at all, we would like to express our gratitude to the reviewers for their great work, since their comments have allowed us not only to significantly improve the manuscript, but also to think about future research.
Point 1: The most significant problem is that the manuscript ignores the issue of C&D reuse and recycling. I understand that this is not the focus of the research; however many cities have had great success in reducing C&D waste through recycling, and mandated use of building techniques that limit waste generation. The authors should at least acknowledge this policy possibility as it would significantly reduce the need for c&d disposal and thus reduce the need for new C&D disposal infrastructure.
Response 1: The section called “3.2 final considerations” was added, which briefly addresses the regulatory context of the study area. See lines 383 to 400. Page 17.
Point 2: Though the comparison of the MCE and network methods was interesting, the authors have limited their inputs, perhaps too much. The highly technocratic approach fails to consider impacts on neighboring residents or indirect environmental impacts, focusing only of the efficiency of transport and disposal, direct environmental impacts, and conditions relevant to waste facilities themselves. Also, beyond acknowledging conservation areas where development is prohibited, the authors have not acknowledged or explored any other regulatory contexts that could be relevant. Though an important investigation, the highly limited and technocratic approach to siting seems a bit outdated, particularly in an environmentally sensitive context like MXCD.
Response 2: The section called “3.2 final considerations” was added, which briefly addresses the regulatory context of the study area. See lines 383 to 400. Page 17.
Point 3: Finally, the authors could do more to generalize their finding. As it is currently written, the article is relevant only to C&D waste policy makers, within MXCD. In order to be published, the authors should consider how the methodological approach or the findings are relevant to policy-makers in other places, or other sectors.
Response 3: Several paragraphs were added in the introduction and conclusions sections, with information highlighting the findings and also the possible work to be done in the future. See lines 64 to 70, page 2. See conclusions section, lines 419 to 426.
Reviewer 2 Report
Overall this work is interesting with a very applied topic in a case-study in Mexico. Although, I believe that it has some potential for minor improvements:
The general description of the problem (Introduction) and the description of its importance for the science and the society could be further improved.
The degree of innovativeness of the methodological approach is not convincingly demonstrated. Some more details about its innovative features could further improve the quality of this paper.Why is this paper likely to be cited in the future?
Please include a bit more text regarding the originality of this work and why it contains new results that significantly advance the research field.
I believe that adding a bit more text on why the results of the method are satisfactory (evaluation approach) will increase the quality of this work. Could the results be more satisfactory if you have changed something in the methodology?
The "Conclusions" section, could be further improved by describing the importance of this work, the highlight of potential further development of this methodology.
Author Response
First at all, we would like to express our gratitude to the reviewers for their great work, since their comments have allowed us not only to significantly improve the manuscript, but also to think about future research.
Point 1: The general description of the problem (Introduction) and the description of its importance for the science and the society could be further improved.
Response 1: Several paragraphs were added in the introduction section, highlighting the importance of the paper. See lines 64 to 70, page 2.
Point 2: The degree of innovativeness of the methodological approach is not convincingly demonstrated. Some more details about its innovative features could further improve the quality of this paper.Why is this paper likely to be cited in the future?
Response 2: Section 3.2 was added, which briefly indicates the contribution of the paper.
Point 3: Please include a bit more text regarding the originality of this work and why it contains new results that significantly advance the research field.
Response 3: See introduction section. See lines 64 to 70, page 2.
Point 4: I believe that adding a bit more text on why the results of the method are satisfactory (evaluation approach) will increase the quality of this work. Could the results be more satisfactory if you have changed something in the methodology?
Response 4: Several paragraphs were added to the introduction sections, results and conclusions, which respond to the reviewer's questions. See highlighted paragraphs of the manuscript
Point 5: The "Conclusions" section, could be further improved by describing the importance of this work, the highlight of potential further development of this methodology.
Response 5: Some paragraphs were added in the conclusions section to highlight the findings. See conclusions section, lines 419 to 426.
Other corrections within the manuscript were also made (highlighted)
- The reference citation in figure 1 was deleted
- 2 new references were added
- A section called Funding was added.
- Some inconsistencies in table 1 were corrected